# OpenReview forum: "SafeAgentBench: A Benchmark for Safe Task Planning of Embodied LLM Agents"
_ICLR.cc/2026/Conference — Submitted to ICLR 2026_

### Official Review · Reviewer_Nig1 · 2025-10-29

**Soundness:** 2
**Presentation:** 2
**Contribution:** 2
**Rating:** 2
**Confidence:** 4

**Summary:**

This paper focuses on evaluating LLM/VLM agents for high-level task reasoning. The paper makes several key contributions:

- A diverse dataset of embodied agent tasks.
- An environment for robotics evaluation.
- Evaluation on state-of-the-art LLM-based planning and reasoning methods.

**Strengths:**

- The paper aims to provide a benchmark and evaluation for robot practitioners to build upon by ensuring VLM-based planners plan safely before executing low-level control policies.
- Assuming you are developing methods assuming an LLM/VLM agent that generates high-level task plans, this work can be useful.
- The plots are well formatted and are clear to read.

**Weaknesses:**

- Narrow Focus on LLM Agents for Embodied Task Reasoning
    - The paper is focused on the use of LLM agents for embodied task reasoning. While this could be useful in some contexts where developers may choose to use an LLM for reasoning, it’s unclear why one would be motivated to do so currently.
    - We’ve seen lots of developments around VLM agents. Why not expand to VLM agents as well? It seems like some methods rely on VLMs, in which case the authors should make this more explicit.
    - Beyond the modalities, the authors need to better motivate how this is a useful problem for roboticists to generate plans from LLMs to control embodied agents.
- Choice of Simulator
    - The reliance on the AI2-THOR simulation environment weakens the extensibility of this benchmark and how useful it is beyond just one simulation environment.
    - This simulator also has a fixed set of 17 high-level actions, which is often a very limited set that undercounts the diverse actions a robot may execute.
- Paper organization and Writing:
    - Early on in the paper, the authors illustrate the central results in Table 1, but there lacks sufficient context to understand the table.
    - Relatively minor point: I don’t like the analogy to the way humans think at the beginning of Section 4.1. This work isn’t motivated by human-type of reasoning, so this comment feels to be a distraction: ‘This controller acts as the agent’s "cerebellum," executing the high-level plans formulated by the planner "brain."’
- Evaluation:
    - It feels very important to clarify that some evaluated methods are open-loop planners and others are closed-loop planners. This makes a difference in the performance, as closed-loop planners will incorporate environmental changes.
    - Table 6 should be referred to in the main text, as it makes it clear what the “abstraction levels” of tasks refer to more precisely.
- Related work:
    - The paper is missing a key related work (in my opinion) in the space of jailbreaking LLM-controlled robots, namely this paper: https://arxiv.org/pdf/2410.13691

**Questions:**

- For the Introduction, how is SafeAgentEnv unique in comparison with other methods?
- Is there a reason why the simulation methods are built around AI2-THOR? Isn’t 17 high-level actions very limiting?
- How does this benchmark environment compare with other benchmark environments commonly used in robotics, such as LIBERO or Bridge?
- Can the authors clarify LLM agents vs VLM agents? See the “Weaknesses” section here.
- Can the authors clarify which planners are open-loop and which ones are closed-loop?
- What about other forms of failure, e.g. collision avoidance in robot planning? Are there tasks developed to evaluate collision avoidance?

---

### Official Review · Reviewer_oNVY · 2025-10-29

**Soundness:** 2
**Presentation:** 3
**Contribution:** 2
**Rating:** 2
**Confidence:** 5

**Summary:**

The paper presents SafeAgentBench, a benchmark designed to evaluate the safety awareness of task-planning embodied LLM agents within an interactive simulation environment (SafeAgentEnv, based on AI2-THOR). The benchmark's key contributions are: (1) A dataset of 750 tasks (450 hazardous, 300 safe) covering 10 hazard types and 3 task structures (detailed, abstract, long-horizon), which notably includes explicitly hazardous instructions often overlooked by prior work. (2) A dual evaluation system using both a rule-based Execution Evaluator and an LLM-based Semantic Evaluator. The authors evaluate 9 agent baselines, primarily driven by GPT-4, and find that their proactive safety awareness is "weak." For instance, the highest rejection rate for detailed hazardous tasks is only 10%. The paper also shows that low risk rates are often due to poor planning capability ($\theta$) rather than intrinsic safety awareness ($\alpha$), and that simple defenses are ineffective.

**Strengths:**

The paper addresses the critical and timely problem of embodied agent safety. As agents become more capable, understanding their failure modes and safety awareness is of paramount importance to the field.

**Weaknesses:**

## Major Weaknesses:

**1.Critically Outdated and Irrelevant Model Selection:**

The paper's experimental setup is fundamentally flawed by its exclusive reliance on pure text-based LLMs.

- **Lack of Multimodality:** Embodied agents, by definition, must perceive and interact with their environment. This requires processing multimodal inputs, primarily visual information (e.g., images, depth maps). The paper’s core evaluation, however, uses GPT-4 (a text-only model) as the central planner, with other text-only models (Llama3, Qwen2, DeepSeekV2.5) as alternatives. This setup is unrealistic. It does not evaluate an embodied agent but rather a disembodied text model's abstract knowledge of safety, which is not the paper's claimed contribution.

- **Outdated Models:** The paper is submitted for 2026 but relies on models from 2023 (GPT-4). The field of LLMs is advancing rapidly. A contemporary evaluation would be expected to include SOTA multimodal models (e.g., GPT-4o, Gemini 1.5/2.5 Pro, Claude 3 family) which are now standard. The paper also fails to investigate whether advanced reasoning techniques (e.g., test-time scaling, complex reasoning methods) could enhance the safety awareness of these models.

**2. Complete Omission of Vision-Language-Action (VLA) Models:**

This is perhaps the most significant flaw. The field of embodied AI has largely moved towards Vision-Language-Action (VLA) models, which are specifically pre-trained or fine-tuned on robotic and physical interaction data. The paper's central claim is that "embodied LLM agents" have poor safety awareness. However, it fails to evaluate any of the models actually designed for this purpose (e.g., RT-2, PaLM-E, or other relevant VLAs). It is entirely possible that VLAs, having been fine-tuned on physical-world data, possess a much stronger "common sense" for physical safety than general-purpose text LLMs. The paper's conclusions are thus an overgeneralization based on the wrong category of models.

**3. Lack of Real-World Validation (sim2real):**

For a benchmark focused on physical safety, a simulation-only evaluation is insufficient. The sim2real gap is a well-known and critical challenge in robotics. Hazards, object interactions, and perceptual failures manifest very differently in the real world than in a clean simulator like AI2-THOR. A convincing paper on this topic must, at a minimum, include a discussion of these limitations or, ideally, provide some preliminary sim2real experiments to validate that the findings in simulation translate to real-world physical safety risks.

**4.Weak and Poorly Designed Defense Strategy Evaluation:**

The paper's attempt to explore defenses (Section 5.6) is superficial and misses the most obvious baseline.

- The authors explicitly state they "do not add any explicit/implicit hint or prompts about safety in any baselines." While this is acceptable for the initial evaluation, it is an unacceptable omission in the defense section.

- The most basic and widely-used defense for LLMs is safety prompting (i.e., adding hints in the system prompt). The authors illogically ignore this baseline and instead test two other methods (CoT filtering and module composition). By failing to compare against the most obvious defense, the conclusions drawn from this section are weak and uninformative.

**5.Limited Scenario Diversity:**


The dataset's scope is narrow, focusing almost exclusively on housekeeping or domestic (kitchen, living room) scenarios. Physical safety is a broad concern that extends to many other environments, such as industrial settings, outdoor navigation, or medical assistance. The benchmark's generalizability is questionable as it omits these other critical domains.



## Minor Weaknesses:

1. Clarity of Metrics: The paper introduces several metric abbreviations in its tables (e.g., Table 2) such as "Rej", "RR(goal)", "RR(LLM)", "ER", and "C-Safe". These abbreviations are not clearly and explicitly defined in the main text, forcing the reviewer to piece together their meanings from context, which hinders readability.

2. Undefined Variables: The variable $\alpha$, introduced on line 371 to represent "intrinsic safety awareness," is used without a clear, self-contained mathematical or conceptual definition, making the subsequent parameter decomposition difficult to follow.

**Questions:**

see above.

---

### Official Review · Reviewer_57jZ · 2025-10-30

**Soundness:** 3
**Presentation:** 3
**Contribution:** 2
**Rating:** 4
**Confidence:** 3

**Summary:**

This paper presents a benchmark designed to evaluate the safety awareness of embodied LLM agents in task planning and execution. The benchmark includes a set of executable tasks including various hazard types and abstraction levels, supported by a simulated environment for testing safety-aware planning behaviors. Experiments show that current embodied LLM agents have limited proactive safety awareness and often fail to identify or reject unsafe instructions. Two baseline defense methods were tested but proved only marginally effective. The paper’s main contribution is providing a benchmark and environment to analyze how embodied LLM agents manage safety risks in planning.

**Strengths:**

1. The paper addresses a highly relevant and timely topic, focusing on the safety of embodied AI systems at a time when LLM-based robotic task planning is rapidly expanding.
2. The benchmark fills an existing gap by shifting the focus from task completion to evaluating how agents respond to hazardous instructions.
3. The work provides a useful starting point for further research on safety evaluation and benchmarking for embodied LLM agents.
4. The design of SafeAgentBench is well thought out, incorporating both explicit and implicit hazards and covering ten risk categories and three abstraction levels.
5. Including multiple LLM backbones makes the results more general and less model-dependent.
6. The paper is well organized and easy to follow, with figures and tables that effectively illustrate the framework, task taxonomy, and experimental results.
7. The semantic evaluation approach, validated through a user study, adds credibility to the evaluation process and complements execution-based evaluation.

**Weaknesses:**

1. The overall contribution feels incremental since the paper focuses mainly on dataset and benchmark construction rather than proposing new algorithms or methods that enhance safety.
2. SafeAgentEnv adds limited novelty because it is largely an adaptation of AI2-THOR with only minor extensions.
3. The evaluation is conducted entirely in simulation, and the paper does not discuss how the findings would transfer to real-world or physical robot scenarios.
4. Mainly reliance on GPT-4 for both dataset generation and semantic evaluation introduces potential model bias, and the rationale for this dependence is not sufficiently justified.
5. The dataset size and number of tested agents may be too small to claim generality or broad representativeness of the benchmark.
6. The experiments on defense strategies are shallow and do not provide strong insight into how to improve agent safety.
7. The description of the Chain-of-Thought safety filter and its implementation details are vague and difficult to understand.
8. Some inconsistencies exist between the text, figures, and supplementary materials, such as differences between results presented with Gemini and DeepSeek.
9. Several research questions are not fully supported with detailed evidence, and the paper could better articulate the novelty and impact of its contributions.
10. The benchmark does not evaluate generalization to new domains such as factory or outdoor environments, which limits its applicability.
11. The dependence on simulated results without deeper statistical validation leaves uncertainty about robustness and reproducibility.

**Questions:**

1. What results would the authors expect if newer models such as GPT-4.5 or GPT-5 were used?
2. Could the authors clarify the "Detailed GP Steps" mentioned in Table 1?
3. Would similar safety patterns appear in industrial or outdoor environments rather than domestic tasks?
4. In Figure 2a, what do the percentages represent?
5. Are all five harm types practically plausible in real robotic systems?
6. How were the unsafe and safe instructions in Table 5 designed and validated?
7. The description of the low-level controller in Lines 255-257 is unclear “Considering that VLA- or RL-trained controllers have not been available, our controller maps each of the 17 high-level actions into sequences of executable low-level APIs, incorporating feasibility checks such as verifying object reachability before execution”. Could the authors elaborate?
8. Should KARMA also appear in Figure 1?
9. Line 361 says 5 agents rejected none. Is this number correct?
10. Was Gemini-2.5-pro evaluated on hazardous tasks?
11. Who participated in the user study that produced 1,008 human ratings? Were they domain experts or general users?
12. How important is safe task planning in simulation compared to real-world deployment?
13. Will the dataset be publicly released?
14. Is the dataset size of 750 tasks sufficient to capture diversity? Can you quantify its linguistic and behavioral variety?
15. Why do abstract tasks include four related instructions?
16. Why are there exactly 17 high-level actions, and are they comprehensive enough?
17. Can SafeAgentBench be adapted to other simulators such as Habitat or IsaacSim, and what modifications would be needed?
18. Have the authors tried fine-tuning LLMs on SafeAgentBench to see if safety awareness improves?
19. When GPT-4 and human evaluators disagreed in semantic evaluation, what types of tasks caused the difference?
20. Could the use of GPT-4 for dataset generation have introduced bias that favors GPT-4-based agents?

---

### Meta-Review · Area_Chair_XkBC · 2025-12-19

**Summary:**

The authors did not submit a rebuttal addressing the concerns raised by the reviewers during the review period. So, the issues identified in the reviews remain unaddressed. After considering the reviewers’ evaluations and the absence of a rebuttal, the Area Chair aligns with the reviewers’ recommendation and recommends rejection.

**Reviewer Concerns:**

No rebuttal was submitted to address the concerns and questions.

**Reviewer Scores:**

No change in the scores since there was no rebuttal.

---

### Decision · Program_Chairs · 2026-01-26

Reject